# The Role of Quarantine on Post-Mortem Performances of Charolaise Young Bulls

**DOI:** 10.3390/ani12040425

**Published:** 2022-02-10

**Authors:** Alessia Diana, Matteo Santinello, Massimo De Marchi, Erika Pellattiero, Mauro Penasa

**Affiliations:** 1Department of Agronomy, Food, Natural Resources, Animals and Environment, University of Padova, 35020 Legnaro, Italy; alessiadiana84@gmail.com (A.D.); matteo.santinello@phd.unipd.it (M.S.); mauro.penasa@unipd.it (M.P.); 2Department of Animal Medicine, Production and Health, University of Padova, 35020 Legnaro, Italy; erika.pellattiero@unipd.it

**Keywords:** biosecurity measure, carcass weight, bovine respiratory disease, antimicrobial resistance

## Abstract

**Simple Summary:**

Despite the positive role of biosecurity measures on the health and productivity of livestock animals, such measures are still poorly implemented in the beef sector due to the high investments that farmers have to face and a lack of empirical information on their effectiveness, specifically with regards to the long-term effects on animal performance and meat quality. Hence, with this short communication we aimed to evaluate the effect of quarantine on post-mortem traits of young beef bulls. Overall, post-mortem performances were positively affected by the quarantine, likely due to a reduced risk of exposure to pathogens that made the animals less susceptible to diseases such as the bovine respiratory disease. In fact, a significant decrease of antimicrobial (AM) use was reported. Implementing the quarantine on-farm can result in an increase of farm profit due to extra sale weight, an amelioration of carcass quality, and an additional saving of AM costs. Additionally, the reduction of AM use is an advantage for both human and animal health given its likely contribution to the development of antimicrobial resistance.

**Abstract:**

Biosecurity is essential to prevent the spread of diseases in livestock animals such as the bovine respiratory disease which can lead to negative effects on growth performance, and carcass and meat quality, as well as to an increase of antimicrobial use. The aim of this study was to investigate the effects of the quarantine on post-mortem performances (hot carcass weight, dressing percentage, pH 60 min after slaughtering, carcass SEUROP conformation, and fat cover) of 576 Charolaise young bulls imported from France and fattened in specialized Italian farms. Approximately half of the animals followed a standard fattening procedure without initial quarantine upon arrival to Italy (NO-QUA) and the other half underwent a 30-day period of quarantine (QUA) since their arrival to the farm. Post-mortem performances and parenteral administration of antimicrobial use were recorded. NO-QUA animals had lower average daily gain and slaughter weight but scored higher for carcass SEUROP evaluation and pH than QUA animals (*p* < 0.05). NO-QUA received more than double the number of parenteral antimicrobial treatments than QUA animals for respiratory diseases (139 vs. 56). Overall, quarantine applied on-farm contributes to improve post-mortem performances while reducing antimicrobial use in beef production.

## 1. Introduction

Morbidity rate and growth performance are important indicators of animal health and welfare with a consequent significant impact on the overall farm productivity [1]. Indeed, animal weight loss and high incidence of diseases are associated with economic losses, since sick or slow growing animals imply a cost in relation to investments in housing, animals purchased, treatments, feeding, and labor [2]. The most significant health issues in beef production are locomotor disorders and the bovine respiratory disease (BRD) [3]. The latter is known for its negative impact on farm profitability due to the reduction of animal performance and the increase of morbidity and mortality rate. Specifically, BRD accounts for nearly 60% to 90% of the cases occurring on-farm [4,5] where a greater exposure to infected animals may increase the risk of transmission [6]. 

Besides the negative effects that BRD can have on the overall performance [7], this disease may also negatively affect meat quality traits such as marbling, fat thickness, meat tenderness, and meat color [8,9]. Moreover, BRD is associated with an increase of antimicrobial use (AMU) and severity of lung lesions at slaughter inspection [7]. Another indicator of meat quality is the pH which, if not appropriate (i.e., beef meat with pH > 5.8–6.0 24 h after slaughter) [10,11], may have negative repercussions on meat color, texture, and palatability with consequent economic losses [12]. Low quality and/or inadequate nutrition [13] as well as stressors—such as seasonal changes, exposure to a new environment, social disruption of the animals, and transportation [14,15]—are recognized as potential factors affecting the pH of beef meat.

BRD can be properly addressed by reducing the risk of exposure of the animals in the herd [16]. An efficient tool to obtain this goal is the application of biosecurity practices, whose contribution in preventing the introduction and spread of pathogens on herd is well-known [17]. Specifically, studies on farm species have already confirmed the positive link between improved biosecurity, animal health, and productivity [18], thus highlighting the importance of biosecurity programs for an efficient farm management. Biosecurity measures identified as valuable in minimizing the risk of diseases in beef production [19,20] are for instance the reduced stocking rate, the application of a period of quarantine to the animals imported and the separation of younger calves from older animals [21,22]. Nonetheless, such measures are still poorly implemented due to the high investments that beef farmers have to sustain and the shortage of studies in beef production aimed at demonstrating their long-term effects on animal performance, meat quality, and economic aspects [23]. Providing evidence of a positive outcome for the beef industry, especially with regards to post-mortem performances, will likely make the farmers less reluctant in adopting such measures on their farms. Hence, the objective of this study was to evaluate the effect of quarantine on post-mortem performances of imported Charolaise young bulls fattened in specialized Italian farms.

## 2. Materials and Methods

### 2.1. Experimental Design

This study was approved by the Ethical Committee for the Care and Use of Experimental Animals of the University of Padova, Italy (Approval No. 74/2018) and was conducted in compliance with Italian law (Decreto Legislativo No. 26/2014) and EU Directive 2010/63/EU on the protection of animals used for scientific purposes. 

Data are part of a larger study carried out in five commercial specialized beef fattening farms located in Veneto region (Italy) and associated to a cooperative of beef producers (AZoVe, Cittadella, Italy) which followed an experimental design as described by Santinello et al. [24]. About 70% of beef cattle produced in Italy are reared in specialized fattening farms in the north-east of the country and approximately 90% of them are young animals imported from France [25]. In fact, this beef production system is characterized by animals reared at pasture in France until 10–14 months of age and then transferred to specific collection centers where are mixed with animals of other farms located in different French departments to create homogeneous batches according to body weight (BW), breed, and sex. Animals are then purchased by Italian beef fatteners and transported to Italy where they reach the slaughter weight after 6–7 months from arrival.

In the present study, only the two male-rearing commercial farms were considered. Briefly, 576 Charolaise young bulls were imported from France between July 2018 and October 2019. Within this timespan, animals arrived in five periods to the two fattening farms at an average body weight (BW0) of 403.37 ± 19.05 kg. At their arrival to the farm, animals were weighed and divided in two experimental groups which were allocated in two different buildings of the farm. Approximately half of the animals of each period was allocated to a control group (NO-QUA, *n* = 264) which followed the standard fattening cycle without initial quarantine, and the other half was assigned to an experimental group (QUA, *n* = 312) which followed a 30-day period of quarantine before moving to the same building of NO-QUA group to follow the remaining standard fattening cycle. The animals of both groups did not receive any vaccinations nor antimicrobial treatments in France, whereas at their arrival to the Italian fattening farms they went through the same vaccination (i.e., polyvalent vaccine for BRD) and parasitic control program and were fed with the same diet. More details regarding animal management can be retrieved from Santinello et al. [17]. Information on the duration of the fattening cycle, BW 30 days after arrival to the fattening farm (BW30), and BW at the end of the fattening cycle (BWfinal) was available for each animal. The number of antimicrobial (AM) parenteral treatments administered to the animals and the reason for their administration (e.g., BRD, lameness, abscess) were also recorded and grouped into three categories—namely respiratory diseases, locomotor disorders and other. The season of arrival at the fattening farm (spring: March, April, May; summer: June, July, August; autumn: September, October, November; winter: December, January, February), the average daily gain from arrival to day 30 of fattening cycle (ADG30, kg/d), the ADG from day 31 to the end of the fattening cycle (ADG31-end, kg/d) and the ADG of the whole fattening cycle (ADGtot, kg/d) were calculated.

### 2.2. Post-Mortem Data

The hot carcass weight was recorded between 45 and 60 min after slaughter and the dressing percentage was calculated as: (hot carcass weight/BWfinal) × 100. The conformation (i.e., SEUROP system—from S (superior) to P (poor)) and fat cover (from 1 (low) to 5 (very high)) evaluation of carcasses was performed by equally trained and blinded technicians according to the classes developed by the European Parliament and Council Regulation (EU) No. 1308/2013 for the assessment of beef quality [26]. The pH (pH_60_) of the *M. gracilis* was measured in triplicate 60 min post-mortem on the right half of the carcass using a pH-meter HD2107.2 Delta Ohm (Delta Ohm, Padova, Italy).

### 2.3. Data Analysis

Data were analyzed using SAS 9.4 software (SAS Institute Inc., Cary, NC, USA) with animal as experimental unit. Information of one animal was missing and thus it was removed from the final dataset before statistical investigation. The SEUROP classes were re-classified assigning a number to each letter—including the corresponding lower (–) and upper (+) values, when applicable [27]—in ascending order, where S = 1, S– = 2, E+ = 3, E = 4, E– = 5, U+ = 6, U = 7, U– = 8, R+ = 9, R = 10, R– = 11, O+ = 12, O = 13, O– = 14, P+ = 15, P = 16, P– = 17. Therefore, lower values were indicative of a better status/quality of the carcass conformation. A similar approach was used for fat cover classes that were re-classified as follows: 1– = 1, 1 = 2, 1+ = 3, 2– = 4, 2 = 5, 2+ = 6, 3– = 7, 3 = 8, 3+ = 9, 4– = 10, 4 = 11, 4+ = 12, 5– = 13, 5 = 14, 5+ = 15. Data were tested for normality and descriptive statistics of all performance traits were calculated. Number of parenteral treatments administered per group (QUA and NO-QUA) and according to the reason of administration were also calculated. 

To investigate the effects of the quarantine on post-mortem performances, an ANOVA test was performed using the GLM procedure of SAS. The BW0 was transformed into a categorical variable (three classes: low, medium, and high) according to mean ± 0.5 standard deviations. The linear model included the fixed effects of farm (two levels), quarantine status (QUA and NO-QUA), season of arrival (four levels), BW0 classes (three levels), the interaction between season of arrival and quarantine status and the interaction between farm and quarantine status. Results are presented as least squares means ± standard error. Multiple comparisons of least squares means of the fixed effects were performed through Bonferroni post-hoc test. Statistical significance was set at *p* < 0.05.

## 3. Results

### 3.1. Effects of Quarantine Status, Farm, and Season of Arrival on the Studied Traits

The ADG30, ADG31-end, and ADGtot averaged 1.99, 1.57, and 1.65 kg/d, respectively, and means of BWfinal, hot carcass weight and dressing percentage were 719.60 kg, 440.59 kg and 61.30%, respectively. SEUROP conformation and fat cover had a median of 4.0 (interquartile range = 4–6) and 5.0 (interquartile range = 4–6), respectively. ADG30, ADGtot, and BWfinal were higher in QUA than NO-QUA animals (*p* < 0.05) whereas no significant differences were observed between groups for ADG31-end, hot carcass weight and dressing percentage. NO-QUA animals had higher score for the carcass SEUROP conformation (*p =* 0.0017) and higher pH_60_ than QUA animals (*p =* 0.0005), whereas no significant differences were observed for carcass fat cover between groups (Table 1 and Table 2). Differences between farms were observed for all traits (*p* < 0.05), except for ADG30, dressing percentage and SEUROP conformation. A season of arrival effect was reported for all the investigated traits (*p* < 0.01; Table 3). Least squares means of the interactions between season of arrival and quarantine status and between farm and quarantine status are available as Appendix A.

### 3.2. Number of Parenteral Antimicrobial Treatments by Group and Reason of Administration

A total of 563 parenteral AM treatments were administered to the animals (NO-QUA = 339 vs. QUA = 224). Specifically, a greater number of treatments was administered to NO-QUA than QUA animals for both respiratory diseases (139 vs. 56) and locomotor disorders (186 vs. 154) while both groups had the same number of treatments for other reasons (14 vs. 14).

## 4. Discussion

The objective of this study was to investigate whether applying a 30-day period of quarantine at the arrival to the fattening farm would have affected post-mortem performances of young Charolaise bulls. The cosmopolitan nature of the breed used in the present study (Charolaise) and the type of beef fattening system [24,25], characterized by animals reared at pasture in the first part of their life, followed by a long transportation distance to reach the fattening farm and by intensive fattening conditions, make our findings representative of similar farm realities. 

Studies on other farm species reported the effectiveness of biosecurity measures on animals’ health and performance [28,29] whereas little is still known about cattle production. Here, we provided practical evidence of similar results in beef bulls that went through a period of quarantine and that improved their post-mortem performances resulting in additional 10 kg BW at slaughter and 4 kg carcass weight, albeit the latter was not significant, compared to NO-QUA group. The final heavier weight of QUA animals was also linked to their higher ADG30 and ADGtot, which highlights the importance of the first 30 days of the fattening cycle in defining the overall growth of the animals. During this initial period, animals have to deal with stressors such as change of diet and new environmental conditions which may affect their health and performance, and likely result in higher AMU [30,31]. Therefore, greater attention should be paid during this sensitive period to obtain valuable outcomes also on the long-term.

The effectiveness of biosecurity in dealing with the transmission of pathogens and the reduction of diseases [17] is also supported by our results since less AM treatments were administered to QUA animals compared to their counterpart. This was especially true for respiratory diseases because NO-QUA animals received more than double the number of parenteral treatments administered to QUA. As expected, the quarantine reduced the risk of exposure to pathogens making the animals less susceptible to diseases such as BRD [16] which is commonly linked to a decrease of animal’s appetite and a depression-like status [32], thus contributing to a rise in morbidity, mortality, and reduced performance [4,5]. Results were in agreement with those of Schnyder et al. [33] who observed an increase of AMU in veal farms that did not quarantine the animals upon arrival. Performance loss seems to increase when other disorders such as arthritis and diarrhea are associated with BRD [9,16]. This may contribute to explain why NO-QUA animals had poorer performance and were more treated than QUA animals for locomotor disorders. Nevertheless, such reduction of AM was not as remarkable as for that reported for respiratory diseases in QUA animals, thus emphasizing the role of other factors on the occurrence of locomotor disorders such as a lack of an appropriate flooring system [34]. Indeed, due to the high BW that Charolaise cattle can reach compared to other breeds [25], animals may be more prone to develop issues such as lameness [35]. Therefore, it is clear to suppose that other strategies may be more efficient than the quarantine in reducing disorders related to locomotion in beef cattle and deserve further investigation.

The negative implications of BRD on meat traits are well-known [8,9]. In the current study, animals that did not undergo the quarantine and were at greater risk of being treated for respiratory diseases, had worse SEUROP carcass conformation and higher pH which are indicators of poorer quality in the beef market [9]. Nevertheless, pH values reported in both groups (QUA and NO-QUA) were higher than the average value considered as good indicator of quality (pH < 5.8) [12,14]. It is important to highlight that the latter is associated to a pH collected 24 h to 48 h after slaughter [10,11]. In our study, pH was recorded 60 min post-mortem and this can explain the higher pH values compared to the literature. Indeed, pH naturally declines over time [36]. Specifically, Barbera et al. [37] reported a decrease of pH from 1 h to 24 h in Charolaise breed, in line with our findings. In addition, the animals used in our study were all males. This further contributes to justify the high pH values observed; indeed, Węglarz [14] reported higher values of pH in males, specifically in young bulls, compared to females. Still, this finding may indicate the necessity of better addressing other factors affecting the pH, such as the long transportation distances and the exposure to a new environment [15,25], and the fact that the quarantine per se, albeit effective, is not sufficient to improve the pH in beef cattle. 

Conversely, fat cover was not affected by quarantine. Indeed, other factors such as nutritional, genetic, and management aspects seem to play a major effect on fat cover [38]. Implementing the quarantine on-farm is expected to enhance farm profit thanks to the extra sale weight, an amelioration of carcass quality, and an additional saving of AM costs. Additionally, the reduction of AM is an advantage for both consumers and public health given the likely contribution of AMU to the development of antimicrobial resistance [39].

Finally, both farm and season of arrival were reported as sources of variation of post-mortem performances. Hence, greater attention to management and biosecurity should be paid by applying strategies tailored to the needs of each farm, since this can lead to improved live and carcass weight and avoid a worsening of meat quality. Seasonal changes are known for their effect on pH due to variations in temperature and humidity [14,15], suggesting that major care should be applied during the coldest periods of the year to overcome potential issues.

## 5. Conclusions

This study showed the effectiveness of the strategy of quarantine to enhance post-mortem performances while leading to a reduction of AMU in Charolaise young bulls. Promoting this practice on-farm is expected to contribute to reduce the negative implications of respiratory diseases on animals’ performance, meat quality, and profitability. Nevertheless, given the novelty of the study, further research on other beef breeds and on a larger sample of farms will be useful to support our findings.

## Figures and Tables

**Table 1 animals-12-00425-t001:** Significance of the effects included in the ANOVA of growth and post-mortem performances ^1^ of Charolaise young bulls (*n* = 575).

	Effect											
	Farm		Quarantine		Season of Arrival		Season*Quarantine		Farm*Quarantine		R^2^	RMSE
Trait	F	*p*-Value	F	*p*-Value	F	*p*-Value	F	*p*-Value	F	*p*-Value		
ADG_30_ (kg/d)	0.60	0.4399	9.63	0.0020	15.50	<0.0001	4.40	0.0045	1.56	0.2128	0.13	0.61
ADG_31-end_ (kg/d)	5.34	0.0212	1.91	0.1677	4.50	0.0039	1.27	0.2846	12.58	0.0004	0.06	0.25
ADG_tot_ (kg/d)	5.15	0.0236	5.41	0.0204	9.59	<0.0001	0.48	0.6957	10.96	0.0010	0.08	0.23
BW_final_ (kg)	15.07	0.0001	4.53	0.0338	7.06	0.0001	0.42	0.7387	9.45	0.0022	0.15	45.73
Hot carcass weight (kg)	11.96	0.0006	1.68	0.1952	7.05	0.0001	0.25	0.8604	8.15	0.0045	0.12	30.13
Dressing percentage (%)	0.51	0.4768	2.01	0.1573	8.77	<0.0001	0.76	0.5176	0.04	0.8504	0.09	1.80
pH_60_	11.85	0.0006	12.33	0.0005	12.14	<0.0001	6.12	0.0004	0.01	0.9053	0.21	0.18
SEUROP conformation	0.15	0.7012	9.93	0.0017	9.75	<0.0001	1.75	0.1549	2.45	0.1183	0.09	1.47
Fat cover	28.05	<0.0001	0.40	0.5261	24.84	<0.0001	0.39	0.7596	6.86	0.0091	0.14	1.24

^1^ ADG_30_ = average daily gain from day 1 to day 30 of the fattening cycle; ADG_31-end_ = average daily gain from day 31 to the end of the fattening cycle; ADG_tot_ = average daily gain of the whole fattening cycle; BW_final_ = body weight at the end of the fattening cycle; Hot carcass weight = weight of the carcass after slaughter and after removal of the head, the internal organs, the limbs and the tail; Dressing percentage = the ratio of hot carcass weight to BW_final_ × 100; pH_60_ = pH measured on the *M. gracilis* 60 min post-mortem; SEUROP conformation = development of carcass profiles, in particular the essential parts (round, back, shoulder) according to the EU Parliament and Council Regulation No. 1308/2013 [26]. A number to each letter (and a corresponding lower (–) and upper (+) value, when applicable) was assigned as follows: S = 1, S– = 2, E+ = 3, E = 4, E– = 5, U+ = 6, U = 7, U– = 8, R+ = 9, R = 10, R– = 11, O+ = 12, O = 13, O– = 14, P+ = 15, P = 16, P– = 17; Fat cover = amount of fat on the outside of the carcass and in the thoracic cavity according to the EU Parliament and Council No. 1308/2013 [26]. Fat cover was re-classified as follows: 1– = 1, 1 = 2, 1+ = 3, 2– = 4, 2 = 5, 2+ = 6, 3– = 7, 3 = 8, 3+ = 9, 4– = 10, 4 = 11, 4+ = 12, 5– = 13, 5 = 14, 5+ = 15. R^2^ = coefficient of determination; RMSE = root mean squared error.

**Table 2 animals-12-00425-t002:** Least squares means (LSM) and standard error (SE) of growth and post-mortem performances ^1^ of Charolaise young bulls (*n* = 575) for farm and quarantine status ^2^ effects.

	Farm				Quarantine			
	Farm 1		Farm 2		NO-QUA		QUA	
Trait *	LSM	SE	LSM	SE	LSM	SE	LSM	SE
ADG_30_ (kg/d)	1.97 ^a^	0.04	1.93 ^a^	0.05	1.85 ^a^	0.04	2.05 ^b^	0.04
ADG_31-end_ (kg/d)	1.61 ^a^	0.02	1.55 ^b^	0.02	1.56 ^a^	0.02	1.60 ^a^	0.02
ADG_tot_ (kg/d)	1.68 ^a^	0.02	1.62 ^b^	0.02	1.62 ^a^	0.02	1.68 ^b^	0.02
BW_final_ (kg)	727.36 ^a^	2.93	709.68 ^b^	3.68	713.36 ^a^	3.42	723.68 ^b^	3.45
Hot carcass weight (kg)	446.30 ^a^	1.93	435.91 ^b^	2.42	439.04 ^a^	2.25	443.16 ^a^	2.25
Dressing percentage (%)	61.37 ^a^	0.11	61.50 ^a^	0.14	61.57 ^a^	0.13	61.30 ^a^	0.14
pH_60_	6.55 ^a^	0.12	6.62 ^b^	0.15	6.62 ^a^	0.14	6.55 ^b^	0.14
SEUROP conformation	4.85 ^a^	0.09	4.79 ^a^	0.12	5.07 ^a^	0.11	4.58 ^b^	0.11
Fat cover	5.03 ^a^	0.08	5.68 ^b^	0.10	5.31 ^a^	0.09	5.40 ^a^	0.09

^1^ ADG_30_ = average daily gain from day 1 to day 30 of the fattening cycle; ADG_31-end_ = average daily gain from day 31 to the end of the fattening cycle; ADG_tot_ = average daily gain of the whole fattening cycle; BW_final_ = body weight at the end of the fattening cycle; Hot carcass weight = weight of the carcass after slaughter and after removal of the head, the internal organs, the limbs and the tail; Dressing percentage = the ratio of hot carcass weight to BW_final_ × 100; pH_60_ = pH measured on the *M. gracilis* 60 min post-mortem; SEUROP conformation = development of carcass profiles, and in particular the essential parts (round, back, shoulder) according to the EU Parliament and Council Regulation No. 1308/2013 [26]. A number to each letter (and a corresponding lower (–) and upper (+) value, when applicable) was assigned as follows: S = 1, S– = 2, E+ = 3, E = 4, E– = 5, U+ = 6, U = 7, U– = 8, R+ = 9, R = 10, R– = 11, O+ = 12, O = 13, O– = 14, P+ = 15, P = 16, P– = 17; Fat cover = amount of fat on the outside of the carcass and in the thoracic cavity according to the EU Parliament and Council No. 1308/2013 [26]. Fat cover was re-classified as follows: 1– = 1, 1 = 2, 1+ = 3, 2– = 4, 2 = 5, 2+ = 6, 3– = 7, 3 = 8, 3+ = 9, 4– = 10, 4 = 11, 4+ = 12, 5– = 13, 5 = 14, 5+ = 15. ^2^ NO-QUA = animals which followed the standard procedure of the fattening cycle; QUA = animals which followed a 30-day period of quarantine before moving to the building of the standard fattening pens. * Results of ADG_30_, ADG_tot_ and BW_final_ were retrieved from Santinello et al. [24]. ^a,b^ Means with different superscript letters within trait and effect are significantly different according to Bonferroni post-hoc adjustment (*p* < 0.05).

**Table 3 animals-12-00425-t003:** Least squares means (LSM) and standard error (SE) of growth and post-mortem performances ^1^ of Charolaise young bulls (*n* = 575) for season of arrival effects.

	Season of Arrival							
	Autumn		Winter		Spring		Summer	
Trait *	LSM	SE	LSM	SE	LSM	SE	LSM	SE
ADG_30_ (kg/d)	1.65 ^a^	0.06	1.94 ^abc^	0.09	2.21 ^c^	0.05	2.01 ^b^	0.04
ADG_31-end_ (kg/d)	1.55 ^ab^	0.02	1.64 ^a^	0.04	1.60 ^a^	0.02	1.52 ^b^	0.02
ADG_tot_ (kg/d)	1.58 ^a^	0.02	1.69 ^ab^	0.04	1.72 ^b^	0.02	1.61 ^a^	0.02
BW_final_ (kg)	705.32 ^a^	4.65	726.97 ^ab^	7.12	729.31 ^b^	3.77	712.48 ^a^	3.27
Hot carcass weight (kg)	437.56 ^ab^	3.08	445.64 ^ab^	4.65	447.66 ^b^	2.48	433.54 ^a^	2.15
Dressing percentage (%)	62.03 ^a^	0.18	61.27 ^ab^	0.28	61.54 ^a^	0.15	60.90 ^b^	0.13
pH_60_	6.61 ^ac^	0.02	6.53 ^bc^	0.03	6.67 ^a^	0.01	6.54 ^b^	0.01
SEUROP conformation	4.71 ^a^	0.15	4.87 ^ab^	0.23	4.41 ^a^	0.12	5.29 ^b^	0.11
Fat cover	5.03 ^a^	0.13	5.38 ^a^	0.19	4.97 ^a^	0.10	6.04 ^b^	0.09

^1^ ADG_30_ = average daily gain from day 1 to day 30 of the fattening cycle; ADG_31-end_ = average daily gain from day 31 to the end of the fattening cycle; ADG_tot_ = average daily gain of the whole fattening cycle; BW_final_ = body weight at the end of the fattening cycle; Hot carcass weight = weight of the carcass after slaughter and after removal of the head, internal organs, limbs, and tail; Dressing percentage = the ratio of hot carcass weight to BW_final_ × 100; pH_60_ = pH measured on the *M. gracilis* 60 min post-mortem; SEUROP conformation = development of carcass profiles, and in particular the essential parts (round, back, shoulder) according to the EU Parliament and Council Regulation No. 1308/2013 [26]. A number to each letter (and a corresponding lower (–) and upper (+) value, when applicable) was assigned as follows: S = 1, S– = 2, E+ = 3, E = 4, E– = 5, U+ = 6, U = 7, U– = 8, R+ = 9, R = 10, R– = 11, O+ = 12, O = 13, O– = 14, P+ = 15, P = 16, P– = 17; Fat cover = amount of fat on the outside of the carcass and in the thoracic cavity according to the EU Parliament and Council No. 1308/2013 [26]. Fat cover was re-classified as follows: 1– = 1, 1 = 2, 1+ = 3, 2– = 4, 2 = 5, 2+ = 6, 3– = 7, 3 = 8, 3+ = 9, 4– = 10, 4 = 11, 4+ = 12, 5– = 13, 5 = 14, 5+ = 15. * Results of ADG_30_, ADG_tot_, and BW_final_ were retrieved from Santinello et al. [24]. ^a,b,c^ Means with different superscript letters within trait and effect are significantly different according to Bonferroni post-hoc adjustment (*p* < 0.05).

## Data Availability

The data presented in this study are available on request from the corresponding author. The data are not publicly available due to their sensitive nature.

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
