# Peer review of "The Role of Quarantine on Post-Mortem Performances of Charolaise Young Bulls"

_animals, 2022, doi:10.3390/ani12040425_

Round 1
Reviewer 1 Report
Introduction
I am missing clear motivation in terms of biosecurity and consequences for beef industry...authors didnt use the potential. The possitive is that it is straigthforward.
Material and Methods.
Standard statistical procedure of comparison to sets of data and several fixed effexts. With such type of data, I have to agree with authors that there is not much to do instead. There is missing description of the size of the sector in Italy (import of calves, fattening - subsidies, country of origine of calves), without it is hard to deside weheter the study is representative.
Results.
Limited regarding the input and SAS outputs. It seems that authors used standard set up of glm procedure regarding significance treshold. I would recommend to do some data polishing.
Discussion.
Short and general, followed by conclusions. I would expect more.
Author Response
Reviewers’ comments are listed below, and authors’ responses (AU) are shown beneath each comment. Changes in the marked revised manuscript are highlighted in yellow.
AU: We would like to thank the Editor and the Reviewers for their professional help in reviewing the manuscript.
Reviewer 1
- Introduction
I am missing clear motivation in terms of biosecurity and consequences for beef industry...authors didnt use the potential. The possitive is that it is straigthforward.
AU: We understand the Reviewer’s point of view. However, we would like to highlight that this short communication is part of a larger study that was recently published (Santinello et al. 2022 - https://doi.org/10.3390/ani12010116) where we provided more information and details as for those requested by the Reviewer. We mentioned this aspect in lines 86 and 108 of the revised manuscript. Maintaining the manuscript concise and straigthforward while still providing the main message of our findings, allowed us to be in line with the standards of the short communications published by the Journal which suggest approximately 2000 words for the whole text. Nevertheless, we agree with the Reviewer that adding some extra information will increase the potential of the paper. See lines 62-75 and 87-94 of the manuscript.
- Material and Methods.
Standard statistical procedure of comparison to sets of data and several fixed effexts. With such type of data, I have to agree with authors that there is not much to do instead. There is missing description of the size of the sector in Italy (import of calves, fattening - subsidies, country of origine of calves), without it is hard to deside weheter the study is representative.
AU: As explained in our comment above, further details on the general Italian beef fattening system are provided in our previous publication (Santinello et al. 2022 - https://doi.org/10.3390/ani12010116). With this short communication, we wanted to focus on important findings related to the final stage of the beef production. Indeed, albeit these findings are part of a larger study, they had merit to be presented. Extra information was added to the text as mentioned in our previous comment. See lines 87-94.
- Results.
Limited regarding the input and SAS outputs. It seems that authors used standard set up of glm procedure regarding significance treshold. I would recommend to do some data polishing.
AU: Editing of the variables was carried out to keep those within the range mean ± 3 SD. Therefore, values that did not fall within this range were considered as missing values in the statistical analysis. After editing, the number of records ranged from 479 for pH60 to 553 for ADG30 e carcass weight.
- Discussion.
Short and general, followed by conclusions. I would expect more.
AU: As explained above, we agreed to be in line with the standards of other short communications published by the Journal. Nevertheless, we understand the Reviewer’s point of view and decided to implement this section to improve the paper. See lines 222-226, 252-258, 262-274 and 292-294.
Reviewer 2 Report
A well written piece about the potential benefit of quarantining cattle before they join the main herd.
There are a few aspects I would like to raise.
Your analysis uses categorised values in an ordinal fashion, which is not a problem, but your GLM function assumes normality of the data, please provide proof of the check that the residuals of the regression model are normally distributed. In the same vein, the reporting in table 2 is on means and SE, which is odd for ordinal data. I can accept some 'poetic license' in the originally linear data (weight, pH) but the grading of the carcass is for me a step too far. This needs reporting in median and an IQR as indication of spread.
Also, in the results it is not clear how the SEUROP grading translates in a number, what does '5' mean, and is S+ a '1' or a '0', and what is the P? It would help the reader to include this detail both in the text as well as in the legend of the tables.
These are my main concerns on the overall well constructed work.
Author Response
Authors’ responses are available in the attached file. Changes in the marked revised manuscript are highlighted in yellow.

Reviewer 3 Report
Dear authors,
English language should be enriched. My suggestions for rephrasing are highlighted in the attached file.
In the design of the study it should be clarified :
- The previous health status of the animal and potential medicine or vaccine administration
- How the allocation was conducted ?
- In the SEUROP classification was the researcher blinded?

Author Response
Reviewers’ comments are listed below, and authors’ responses (AU) are shown beneath each comment. Changes in the marked revised manuscript are highlighted in yellow.
AU: We would like to thank the Editor and the Reviewers for their professional help in reviewing the manuscript.
Reviewer 3
- Dear authors, English language should be enriched. My suggestions for rephrasing are highlighted in the attached file.
AU: Amended. See the new version of the manuscript (lines 51-52, 55-60, 86, 109-111 and 114-116).
In the design of the study it should be clarified:
- The previous health status of the animal and potential medicine or vaccine administration
AU: The experimental animals went through the same health protocol. Specifically, they did not receive any vaccinations nor antimicrobial treatments in France. Whereas, at their arrival to the Italian farms involved in the study, the same vaccination programme (i.e., polyvalent vaccine for BRD) and parasitic control programme were administered to the animals.
Also, we would like to highlight that this short communication is part of a larger study that was recently published (Santinello et al. 2022 - https://doi.org/10.3390/ani12010116) where we provided more information and details on the design of the study. We mentioned this aspect in lines 86 and 108 of the revised manuscript. Maintaining the manuscript concise and straigthforward while still providing the main message of our findings, allowed us to be in line with the standards of the short communications published by the Journal. Nevertheless, we agree with the Reviewer that adding few extra information on the health protocol will make the paper more robust. See lines 104-107.
- How the allocation was conducted?
AU: As explained in our comment above, more details on the design of the study and animal management are available in our previous manuscript (Santinello et al. 2022 - https://doi.org/10.3390/ani12010116).
- In the SEUROP classification was the researcher blinded?
AU: Yes, we confirm that the technicians who collected the SEUROP data were blinded. We added this information to the text. See line 123.
Round 2
Reviewer 1 Report
I would thank authors for their response and according revisions made.
With this I dont have any further comments and revisions regarding the manuscript.